# Dynamics of CSBD Healing after Implementation of Dentin and Xenogeneic Bone Biomaterial

**DOI:** 10.3390/ma16041600

**Published:** 2023-02-14

**Authors:** Olga Cvijanović Peloza, Ana Terezija Jerbić Radetić, Mirjana Baričić, Luka Bukovac, Sanja Zoričić Cvek

**Affiliations:** 1Department of Anatomy, Medical Faculty, University of Rijeka, 51000 Rijeka, Croatia; 2Clinical Orthopaedic Hospital Lovran, 51415 Lovran, Croatia; 3Medical Faculty, University of Zagreb, 10000 Zagreb, Croatia

**Keywords:** CBCT, cerabone, dentin, bone regeneration, osteoinduction, TNF-α, OPN, VEGF

## Abstract

Autologous dentin is frequently used in guided bone regeneration due to its osteoinductive properties, which come from its similarity to native bone. On the other hand, the xenogeneic bone biomaterial Cerabone^®^ serves as a biocompatible, but hardly resorbed biomaterial. During bone healing, an inflammatory, vascular, and osteogenic response occurs in which cytokines such as tumor necrosis factor-alpha (TNF-α), vascular endothelial growth factor (VEGF), and osteopontin (OPN) are released locally and systemically. The aim was to follow up the dynamics (on days 3, 7, 15, 21, and 30) of critical-sized bone defect (CSBD) healing after the implantation of bovine devitalized dentin, rat dentin, and xenogeneic bone biomaterial. For this purpose, histological and histomorphometric methods were employed. Additionally, serum concentrations of TNF-α, VEGF, and OPN were monitored in parallel to better understand the biomaterial-dependent systemic response in rats. At the last time interval, the results showed that the bone defect was bridged over in all three groups of biomaterials. The rat dentin group had the highest percentage of bone volume (BV/TV) and the least percentage of residual biomaterial (RB), which makes it the most optimal biomaterial for bone regeneration. Serum concentrations of the TNF-α, VEGF, and OPN refer to systemic response, which could be linked to intense bone remodeling between days 15 and 21 of the bone healing.

## 1. Introduction

Guided bone regeneration (GBR) is a surgical procedure that uses barrier membranes and bone substitute grafts to provide sufficient bone volume for implant placement. In this procedure, natural biomaterials, which can be autologous (autograft), homologous (allograft), and heterologous (xenograft), are applied, as well as synthetic biomaterials [1,2,3].

Nowadays, xenografts are used in oral surgery; they are most often produced from the bones of cattle and pigs. They undergo purification processes, but some, such as Bio-Oss^®^, retain the characteristics of bone, whereas the rest only contain hydroxyapatite structures. There is also the possibility of combining xenografts in order to improve their osteoinductive properties, either with growth factors such as BMP7 or with bioactive substances such as collagen, hyaluronic acid, or fibrin.

Allogeneic grafts, on the other hand, which are originally obtained from decellularized cadaver tissue, have the advantage of retaining bioactive molecules. They come in various forms (powders, gels, fillers) in order to improve the periodontal tissue regeneration. However, regardless of the retention of osteoinductive molecules, they come with a risk of transmissible diseases (e.g., HIV, hepatitis B and C).

Autologous grafts are the gold standard since they avoid immune responses that lead to the rejection of the implanted graft. However, taking a graft from the calvarial or iliac bone often leads to disorders such as infection or deterioration of the donor site. With autologous bone transplants, there is another problem—bone resorption is increased [3].

Recently, autologous dentin has been increasingly used in GBR. The presence of bone morphogenetic proteins and other growth factors in the demineralized dentine matrix contribute to its osteoinductive properties. In addition to the above, the similarity of dentin with bone tissue is also evident from the presence of collagen type I and type III and bone matrix proteins, such as OPN, osteocalcin (OCN), and bone sialoprotein (BSP) [4,5,6]. Several studies on animal bone defect models have applied histological and histomorphometric methods to prove good osteoconductive and osteoinductive properties of dentin [7,8,9,10]. Furthermore, a clinical histomorphometric study referred to 63% of newly formed bone after 7 months of guided bone regeneration with autologous dentin [11]. However, studies that compare demineralized dentin matrixes with xenogeneic biomaterials are rare. One example was a clinical study by Pang et al., who applied Bio-Oss^®^ and dentin in alveolar ridge preservation and found no significant difference in the percentages of a newly formed bone [12].

Bio-Oss^®^ and Cerabone^®^ are among the most commonly used xenografts in dentistry. Even though they share the similarity that both are of bovine origin, these two biomaterials differ in their physical properties. The manufacturer states a particle size of 500–1000 μm for Cerabone^®^, compared with 250–1000 μm for Bio-Oss^®^. Except for the larger particle size, Cerabone^®^ has larger hydroxyapatite crystals and lower porosity than Bio-Oss^®^. Large hydroxyapatite crystals are related to lesser degradability, and lower porosity is related to lesser penetration of the blood vessels and secondarily to lesser osteoblast adhesion [13,14].

Only a few studies have investigated Cerabone^®^ in models of critical-sized bone defects. Huber et al. found 55% of newly formed bone after 60 days of bone defect healing in rabbits [15]. In a study of 8 mm bone defects in rats, Shakir et al. demonstrated a gain in bone density, which increased from 40.81 ± 1.12% to 77.69 ± 0.93% at 4 and 8 weeks, respectively. However, the authors indicated that Cerabone^®^ was not yet degraded at 8 weeks [16]. In the most recent CSBD study of a 5 mm rat calvarial defect, bone volume values of 53.085% were assessed for Cerabone^®^ on the 30th day, along with 10.329% of residual biomaterial [14]. Clinical studies with Cerabone^®^ indicated bone volume gains between 23.40% and 31.63% [13]. During bone healing, conditions of the microenvironment, including inflammation, vascularization, and normal function of osteoblasts, are crucial for the integration of biomaterials and their degradation.

Among various cytokines, TNF-α, VEGF, and OPN stand out as important inflammatory, neoangiogenesis, and mineralized matrix factors in bone regeneration and biology.

TNF-α is one of the crucial inflammatory cytokines involved in bone remodeling that, together with RANKL, promotes osteoclastogenesis and bone resorption [17,18]. There is also evidence of TNF-α as an important inducer of the differentiation of mesenchymal stem cells (MSC) into osteoblasts [14,19,20]. VEGF enhances neovascularization and stimulates vascular endothelial cells to secrete growth factors and cytokines that affect the differentiation of mesenchymal cells towards osteogenic cells [21]. OPN is a non-collagenous protein of the mineralized bone matrix. Its mRNA is expressed in preosteoblasts, but it is mostly secreted by mature osteoblasts at sites of bone remodeling [22].

Given the above, we assume that due to its osteoinductive properties, dentin could contribute to better healing of critical-sized bone defects by stimulating a higher percentage of newly formed bone compared with xenogeneic bone biomaterial.

The aim was to perform a histomorphometric analysis to obtain the values of bone volume and residual biomaterial and compare these between groups of devitalized xenogeneic dentin and rat dentine with a group of xenogeneic bone biomaterial. Additionally, the aim was to monitor serum concentrations of TNF-α, VEGF, and OPN in parallel to better understand the systemic effects of biomaterial-dependent bone regeneration.

## 2. Materials and Methods

### 2.1. Animals and Experimental Design

Wistar 2.5-month-old male rats were used in this study. The animals were randomly divided into three groups, with five animals per group and 25 animals in the control group (Table 1). The first group was implanted with Cerabone^®^ (Botiss Biomaterials GmbH, Berlin, Germany), the second with bovine devitalized dentin (dentine slices, LKB Vertriebs GmbH, Wien, Austria), and the third with allogeneic dentin matrix derived from teeth. Implanted calvarial defects were covered with a collagen membrane (Mucoderm^®^, acellular dermal collagen matrix, Botiss Biomaterials GmbH, Berlin, Germany). The control group included animals with bone defects covered with a collagen membrane.

### 2.2. Surgical Protocol

The surgical protocol was followed to the letter according to the paper published by Jerbić Radetić et al. [12], with a few minor changes.

After administration of ketamine (80 mg/kg, i.p.) and xylazine (5 mg/kg, i.p.) anesthesia, the animals received an intraperitoneal injection of tramadol (10 mg/kg). After that, subcutaneous injection of 0.3–0.4 mL of 1% lidocaine was administered at the incision site. Vital parameters were constantly monitored by a pulse oximeter (Mouse STAT, Pulse Oximeter and Heart Rate Monitor Module, Kent Scientific Corporation, Torrington, CT, USA). The hair was then shaved from the ridge of the muzzle to the caudal end of the calvaria using an electric trimmer specialized for small animals (MOSER 1556 AKKU, professional cordless hair trimmer, BIOSEB In Vivo Research Instruments, Unterkirnach, Germany). A scalpel was used to make the incision into the skin and into the periosteum over the scalp. After lateral contraction, the calvaria was visualized. A defect was made in the frontoparietal complex using a trephine with an outer diameter of 8 mm (Helmut Zepf, Seitingen-Oberflacht, Germany) at 1500 rpm or less. Low trephine speed and wetting with saline were crucial to avoid bone injuries that could interfere with the results. For site standardization, a rat head holder (Model 920-E Rat Head Holder, David Kopf Instruments, Los Angeles, CA, USA) was used along with a tissue-marking instrument (Biopsy Punch, Kai Medical, Tokyo, Japan). The amount of the applied biomaterial was assessed with a precision scale (ME-T Precision Balance, Mettler Toledo, Zagreb, Croatia). In the group in which an allogeneic dentin matrix was used, we performed molar extraction. The crown of the teeth was sawn off, and the rest of the teeth, which mostly consisted of the root, were further processed. Prior to the application, bovine devitalized dentin and rat dentine were ground using a dentin grinder (KometaBio Smart Dentin Grinder, KometaBio, Fort Lee, NJ, USA). The size of the ground biomaterial was 0.5–1 mm, resembling the size of Cerabone**^®^** particles. After the grinding process, dentin was stored in sterile vials, after which a cleanser was added. This contained a mixture of 0.5 M sodium hydroxide and 20% ethanol. Dentin was kept in the solution for 5 min. After rinsing with sterile saline and removing excess fluid with sterile gauze, EDTA was added for 2 min. Again, excess fluid was removed using sterile gauze, and finally, phosphate-buffered saline (PBS) was used for rinsing at least two times. Dentin particles were dried using sterile gauze, after which they were stored in sterile containers and ready for application. The implanted biomaterials were covered using a collagen membrane (Mucoderrm, acellular dermal collagen matrix, Botiss Biomaterials GmbH, Berlin, Germany), after which suturing with simple sutures followed (3-0 USP, Hu-Friendly Perma Sharp Sutures, Hu-Friedy Mfg. Co., Chicago, IL, USA). After the operation, animals were transported in cages with heating pads (Heating pad for rats—20.5 × 12 cm^2^, DC temperature controller, FHC, Bowdoin, ME, USA) to reduce postoperative trauma.

This study was conducted according to the guidelines of the Declaration of Helsinki and approved by the Ethics Committee of the University of Rijeka and the Ministry of Agriculture (EP 302/2021).

### 2.3. Laboratory Processing

After calvarial bone tissue was collected, the samples were stored in a 4% paraformaldehyde solution and kept in a refrigerator at 4 °C. Blood samples were obtained by heart puncture of the left ventricle and centrifuged at 5500 rpm for 15 min. Serum was then aliquoted into plastic containers with caps and stored at −80 °C.

### 2.4. Histological Staining

The bone samples were decalcified for several days using Osteofast 2 (Biognost, Zagreb, Croatia) and fitted into a paraffin block. Later, they were cut using a microtome (Leica, RM2155, Wien, Austria) for the desired thickness (3–5 μm). Tissue sections were stained with histological stain hematoxylin and eosin (HE). The region of interest (ROI) showed the edges of the bone defect and the area in between.

### 2.5. Histomorphometry

After decalcification and fitting into paraffin blocks, samples were cut using a microtome. Multiple serial sections were prepared for the analysis of each tissue sample. Histomorphometry was performed using a light microscope (Olympus BHA, Tokyo, Japan) to which a digital camera (Sony, Yokohama, Japan) was adapted. Magnification was 40×. Microphotographs of histological sections were transferred to the monitor of the computer software for quantitative analysis of the microscopic images (VAMS, Zagreb, Croatia), and the parameter values were quantified as follows: bone volume/total volume, BV/TV (%), and residual biomaterial/total volume, RB (%).

### 2.6. Enzyme-Linked Immunosorbent Assay

Serum concentrations of osteogenic (OPN), inflammatory (TNF-α), and vascular cytokines (VEGF) were quantified in rat serum samples using specific quantitative immunoenzymatic tests, i.e., the ELISA method, carefully following the manufacturer’s instructions (Abcam, Cambridge, United Kingdom) for the individual tested analyte. After the analytical procedures, measurements were made on a microtiter plate reader (Bio-tek EL808, Bio-tek instruments INC, Winooski, VT, USA) by determining light absorption at a wavelength of 450 nm.

The principle of operation of the aforementioned tests is based on the principle of quantitative enzyme immunoassay. Before performing the immunoenzymatic test, all reagents and analytes must be at room temperature, and solutions were prepared immediately before use from the enclosed concentrated solutions or lyophilized reagents. Briefly, wells on an analytical microtiter plate were coated with the appropriate anti-rat monoclonal antibody. Standards and samples were applied in duplicate to pre-defined wells in the appropriate volume and dilution defined for each analyte. Incubation was carried out for a precisely defined period at room temperature, i.e., at a temperature of 37 °C, depending on the protocol, while the wells were covered with adhesive foil and left on a laboratory shaker. After the prescribed period of incubation of the serum sample or standard, the contents of the wells were emptied, and each well was washed four times with 350 μL of washing buffer. The procedure was carried out using a multichannel pipette, and after each washing, the plate was dried well on cellulose cotton wool. After washing, a specific secondary antibody was added. A biotin-labeled polyclonal anti-rat antibody recognizes the target peptide. After that, the biotinylated peptide interacted with the streptavidin-horseradish peroxidase (SA-HRP) complex, which catalyzed the degradation of the chromogenic substrate. The substrate degradation reaction was stopped by adding 100 μL of stop solution to each well. In doing so, a yellow coloration occurred, and the intensity of the resulting color, which needed to be measured within 20 min, was directly proportional to the amount of the determined analyte in the standards or individual tested samples. The obtained light absorbance values of standards of known concentrations were used to create a standard curve and to determine the unknown concentration of the required analyte in rat serum samples. Unknown concentrations in the tested samples were determined from the standard curve equation and expressed in pg/mL. Only OPN was expressed in ng/mL.

### 2.7. Statistical Analyses

Statistical analysis was performed with the help of the computer program Statistica 14.0.1.25 (TIBCO Software Inc., Palo Alto, CA, USA). The normality of the distribution was tested with the Kolmogorov–Smirnov test, which showed that the data were normally distributed. Differences between the groups were tested using the analysis of variance, a repeated measures ANOVA. The Scheffe post hoc was used to determine specific differences between groups. Results were presented as means ± SD and were considered statistically significant at *p* < 0.01.

## 3. Results

### 3.1. Histological Analyses

On day 3, an organized blood clot with inflammatory cells at the top of the bone defect of the Cerabone group and control was observed (Figure 1a,j). At the same time interval, all defect sites were filled with particles of biomaterials and loose connective tissue (Figure 1a,d,g). On day 15, in the Cerabone group and dentine representatives groups, biomaterial particles occupied the space of the central part of the defect, which was bridged over by the fibrous tissue. Newly formed bone was visible along the edges of the bone defect and in small amounts around the biomaterial particles (Figure 1b,e,h). On day 30, woven bone spread from the edges towards the central part of the bone defect. In the dentine and Cerabone groups, the particles of residual biomaterial were visible above the bony bridge formed by the cortical bone (Figure 1c,f). In the rat dentin group, no biomaterial particles were observed, and the bone defect was bridged over by the lamellar bone on the periphery and by woven bone at the central area (Figure 1i). The calvarial defect of the control group was mainly filled with fibrous tissue at all time points. These observations suggest that the effect of biomaterials on bone regeneration is restricted to the area of the biomaterial particles. The newly formed bone at the edge of the defect is a result of the intrinsic regenerative capacity of the bone tissue.

### 3.2. Histomorphometric Analyses

Mean values ± SD of the bone volume (BV/TV) and residual biomaterial (RB) are presented in Figure 2. The results point to statistically significant variations in the BV/TV by days and by biomaterials. The RB showed statistically significant variations by days. A linear increase in the BV/TV was observed in all three groups of biomaterials and the control group, and RB linearly decreased (Figure 2).

#### 3.2.1. Variations in the Bone Volume (BV/TV, %) by Days and within Set Time Points

On days 21 and 30, the Cerabone group and control had significantly higher values than on day 3 (*p* < 0.01). During the last three time points, both representatives of dentine groups showed significantly higher values than on day 3 (*p* < 0.01). On day 30, the Cerabone group had significantly higher values than on day 7 (*p* < 0.01). At the last two time points, both representatives of dentine groups and the control showed significantly higher values than on day 7 (*p* < 0.01). At the last time point, both representatives of dentine groups showed significantly higher values than on day 15 (*p* < 0.01) (Figure 2a).

During the last two time points, the rat dentin group showed significantly higher values than the Cerabone group (*p* < 0.01). On day 30, the Dentin group showed significantly higher values than the control (*p* < 0.01). On the other hand, the rat dentin group showed significantly higher values than the control at the last two time points (*p* < 0.01) (Figure 2a).

#### 3.2.2. Variations in the Residual Biomaterial (RB, %) by Days

On day 3, all biomaterial representatives showed significantly higher values than on day 30 (*p* < 0.01). During the following time point, on day 7, both representatives of dentine groups had significantly higher values than on day 30 (*p* < 0.01). On day 15, the dentin group had significantly higher values than on day 30 (*p* < 0.01) (Figure 2b).

### 3.3. Analyses of the Serum Concentrations

Serum concentrations of the TNF-α, OPN, and VEGF are presented in Figure 3. The results indicate statistically significant variations in the given cytokines by days and by biomaterials.

#### 3.3.1. Variations in the Serum Cytokine Concentrations by Days and within Set Time Points

##### Tumor Necrosis Factor-Alpha (TNF-α, pg/mL)

On day 3, the dentine group had significantly higher values than on days 15 and 30 (*p* < 0.01). On days 7 and 21, the Rat dentine group showed significantly higher values than on day 3 (*p* < 0.01). On day 7, the Cerabone group showed significantly higher values than on days 15, 21, and 30 (*p* < 0.01). The rat dentine group showed significantly higher values on day 7 than at the last time point (*p* < 0.01). On the other hand, the Cerabone group showed the same trend during day 15 as opposed to day 30 (*p* < 0.01). The rat dentine group showed significantly higher values on day 21 as opposed to days 15 and 30 (*p* < 0.01) (Figure 3a).

On day 3, the Cerabone group and dentin group had significantly higher values than the rat dentin group and the control (*p* < 0.01). On day 3, the rat dentin group had significantly higher values than the control (*p* < 0.01). On day 7, the Cerabone group had significantly higher values than the dentin group, rat dentin group, and control, and during the same time point, both representatives of dentin groups had significantly higher values than the control (*p* < 0.01). When comparing the two dentine representatives, significantly higher values were found in the rat dentin group as opposed to the dentin group (*p* < 0.01). On day 15, the Cerabone group and the rat dentin group had significantly higher values than the dentin group (*p* < 0.01). All the representatives of the biomaterial groups showed significantly higher values than the control on days 15 and 21 (*p* < 0.01). On day 21, the rat dentin group showed significantly higher values than the dentin group (*p* < 0.01). On day 30, the Cerabone group, dentin group, and rat dentin group showed significantly higher values than the control (*p* < 0.01). On the same day, the Cerabone group showed significantly higher values than the dentin group and the eat dentin group (*p* < 0.01) (Figure 3a).

##### Osteopontin (OPN, ng/mL)

The dentine group had significantly higher values on day 21 than at any other time point (*p* < 0.01).

On day 21, both representatives of dentin groups showed significantly higher values than the Cerabone group, and on the same day, the dentin group also showed significantly higher values than the control (*p* < 0.01) (Figure 3b).

##### Vascular Endothelial Growth Factor (VEGF, pg/mL)

The Cerabone group had significantly higher values on days 7 and 15 as opposed to day 3 (*p* < 0.01). At days 3 and 7, the Cerabone group showed significantly higher values than at the last two time points (*p* < 0.01). On days 7 and 15, the dentine group had significantly higher values than on day 3 (*p* < 0.01). On days 7, 15, and 30, the rat dentine group had significantly higher values than on day 3 (*p* < 0.01). On day 7, the control had significantly higher values than on day 3 (*p* < 0.01). The dentin group had significantly higher values on day 15 than on day 7 (*p* < 0.01). On the other hand, the dentin group on day 7 had significantly higher values than on days 21 and 30 (*p* < 0.01). The rat dentin group had significantly higher values on day 15 than on day 7, and the same group showed significantly higher values on day 7 as opposed to day 21 (*p* < 0.01). The control had significantly higher values on day 7 as opposed to the last two time points (*p* < 0.01). All biomaterial groups showed significantly higher values on day 15 as opposed to the last two time points (*p* < 0.01). The rat dentin group showed significantly higher values on day 30 in comparison to day 21 (*p* < 0.01) (Figure 3c).

During the first two time points, the Cerabone group and the dentin group had significantly higher values than the rat dentin group and control (*p* < 0.01). On day 15, all the representatives of the biomaterial groups showed significantly higher values than the control (*p* < 0.01). In addition, on the same day, the dentin group showed significantly higher values than the Cerabone group and the rat dentin group (*p* < 0.01). On day 21, the dentin group showed significantly higher values than the control (*p* < 0.01). During the last time point, the rat dentin group showed significantly higher values than the Cerabone group, the tdentin group, andhe control (*p* < 0.01) (Figure 3c).

## 4. Discussion

The general trend showed that BV/TV values increased with time intervals, whereas RB values decreased. Analyses within set time points revealed higher values of BV/TV in the rat dentin group compared with the Cerabone and control groups in the last two time intervals. Furthermore, the dentin group had higher values than the control at the last time interval.

Autologous dentin in guided bone regeneration comes as a natural choice considering that it has the most similar chemical composition of organic and inorganic substances to bone tissue. Rat dentin used in this study served as an allograft, but it could also be considered an autograft given the genetic similarity between Wistar rats.

The calvarial bone is not so commonly used in maxillofacial surgery. Applying the calvarial bone graft was described in the case of severe atrophy of the mandible, which resulted in minimal bone reduction ranging from 0.83% to 5.83% after implant loading, with a 100% implant survival rate after a one-year follow-up. On the other hand, the iliac crest graft is susceptible to excessive bone resorption, which can lead to bone volume reduction from 47% to 49.5% and subsequently to loss of dental implants. The latter suggests that much better results are obtained when a bone graft of the same embryonic origin is used in craniofacial prosthetics [23].

In a study conducted on an experimental model, surgical bone defects were made in 36 adult rabbits divided into four groups: bone defect (control), bone defect covered with membrane, bone defect with dentin, and bone defect with dentin and membrane. After rabbits were sacrificed at 30, 60, and 90 days, histological and histomorphometric analyses were performed on the bone samples. The results of the 30th day showed an increase in the newly formed bone of 54.20 ± 1.23 in the group of the homogenous demineralized dentin matrix with membrane [7]. This is comparable with our results of the histomorphometric analyses at the same time interval, with 35.817% of bone volume in the dentin group and 39.675% in the rat dentin group (Figure 2). In another study, an autologous demineralized dentin matrix was used in a 10 mm calvarial defect in mini pigs, which resulted in extensive bone formation at 4 weeks, and the control group presented with fibrosis [8]. This result also supports our findings of newly formed bone in both representatives of the dentine groups and the control, out of which the latter was characterized by fibrous tissue in the middle of the bone defect (Figure 1k,l).

In the range of literature reports that compared dentin with other biomaterials, Bio-Oss^®^ had and still holds the primacy. BioOss is a well-known bovine xenograft whose experimental history stretches back 25 years. It was used in the treatment of periodontal osteogenic defects, guided bone regeneration, and sinus augmentation procedures [24]. In this study, we used Cerabone^®^, which also has a wide range of applications in GBR. During the last time interval, the lowest BV/TV values and the highest RB values were found in the Cerabone group. Such a result goes along with a clinical study report of a long-term presence of Cerabone^®^ after sinus augmentation, implying that Cerabone^®^ is hardly resorbed biomaterial [13]. Although Cerabone^®^ provides an osteoconductive model for bone formation, it does not have any osteoinductive characteristics, which may mean adding growth factors to improve bone formation in defects. One of the rare studies that compared granulated dentin and xenogeneic biomaterial is that of Habashneh et al., who investigated the rate of a 5 mm rat calvarial defect closure without quantifying bone parameters. The authors discovered that the best closure of the largest diameter was found in the group that used a mixture of dentin and Bio-Oss^®^, followed by the group with dentin. In terms of bone formation, all examined groups showed significant bone formation in the eighth week compared with the control group. However, there were no statistically significant differences between the groups with dentin, Bio-Oss^®^, or a mixture of the two in terms of reduction in the largest diameter. As the study used human dentin with human proteins, the comparison of bone formation was overshadowed by the immune response generated by human proteins in rats [10].

According to our results, bridging of the defect with cortical bone was observed in the last time interval (Figure 1c,f,i). It is clear that the calvarial defect closed much earlier than in the above-mentioned study, and that the residual biomaterial was completely incorporated into the newly formed bone. Given that we used two dentin biomaterials—bovine devitalized dentin and rat dentin, we saw the following differences: bovine devitalized dentin was anchored in the newly created bone and retained more residual biomaterial, whereas rat dentin was also anchored but with a lower percentage of residual biomaterial, which implies its better resorption (Figure 1f,i).

Our results revealed that systemic response in groups of biomaterials significantly differed from the control. The latter was observed for all three cytokines. Regarding TNF-α, two increases were found. The first was on day 3 when Cerabone and dentin groups showed significantly higher TNF-α concentrations than the rat dentin and control groups. On day 7, higher concentrations of TNF-α persisted in the Cerabone group, which significantly differed from both representatives of dentine groups and the control. Such a result implies a strong immunogenic response, which can be explained by the recipient’s reaction to the xenogeneic biomaterial. The second increase in TNF-α concentrations was found on day 21, when all three groups of biomaterials significantly differed from the control, and the rat dentin group differed from the dentin group (Figure 3a). During bone healing, inflammatory cells secrete numerous proinflammatory cytokines such as TNF-α, interleukine-1 (IL-1), and interleukine-6 (IL-6). TNF-α helps in the recruitment of mesenchymal stem cells and induces osteoclastic function [14,17,18,19,20]. This is important at the early stages of bone healing, but also at the late stages when the woven bone is remodeled into lamellar bone. We assume that the second increase in TNF-α on day 21 was related to intensified bone remodeling, which, in terms of bone resorption, contributed to the least percentage of RB in the rat dentine group.

On day 15, all three biomaterials showed a significant increase in the VEGF, which differed from the control. The dentin group also showed significantly higher values than the rat dentin and Cerabone groups (Figure 3c). VEGF concentrations indicate that new blood vessels are formed, which will bring precursors of the bone cells to the bone defect site. Furthermore, VEGF helps bone healing by inducing the proliferation of mesenchymal cells into osteoblasts [21].

Along with TNF-α, OPN increased on day 21, which was especially pronounced in representatives of the dentine groups. Namely, in the mentioned time interval, the dentin and rat dentin groups had significantly higher values than the Cerabone group, and the dentin group had significantly higher values than the control (Figure 3b). Such OPN concentrations could be related to osteoblasts activity during bone formation or can speak for its increased release in systemic circulation due to biomaterial resorption.

## 5. Conclusions

All three biomaterials induced new bone formation, which bridged over critically sized bone defects in rats. When compared, the optimal biomaterial in bone regeneration is rat dentin, as it was proved by the highest percentage of bone volume and the least percentage of residual biomaterial. Considering that the highest percentage of residual biomaterial was found in the Cerabone group, it can be concluded that it is not an easily resorbable biomaterial. The extent of regeneration of a critically sized calvarial defect in rats could be monitored by the use of systemic cytokines. At the time interval between days 15 and 21, the most intensified bone remodeling occurred, as it was demonstrated by serum concentrations of TNF-α, VEGF, and OPN.

## Figures and Tables

**Figure 1 materials-16-01600-f001:**
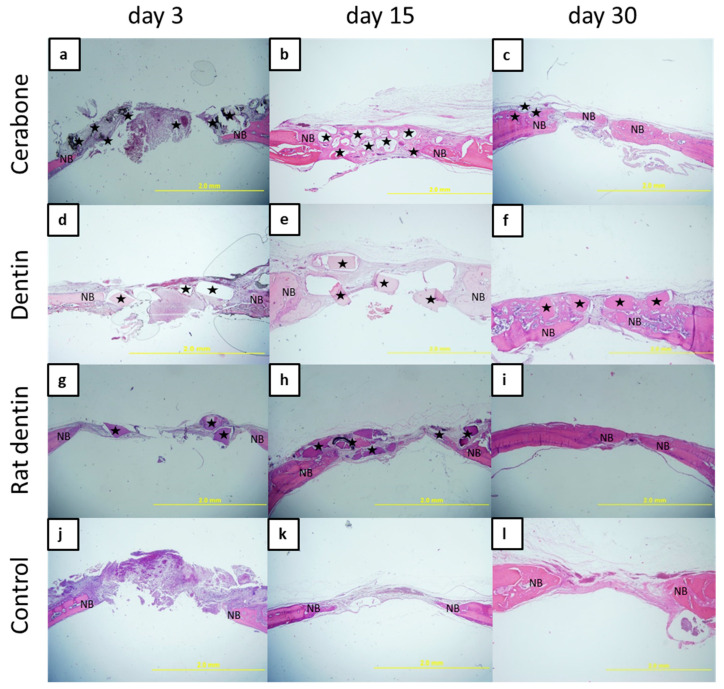
Microphotographs of the coronal sections of rat calvarial bone defects with implanted biomaterials: Cerabone (**a**–**c**); dentin (**d**–**f**); rat dentin (**g**–**i**), and control (**j**–**l**). For each group, three representative time points were chosen: 3 days (**a**,**d**,**g**,**j**), 15 days (**b**,**e**,**h**,**k**), and 30 days (**c**,**f**,**i**,**l**). In each tissue section, biomaterial (★) and newly formed bone (NB) were marked (HE staining, magnification 40×).

**Figure 2 materials-16-01600-f002:**
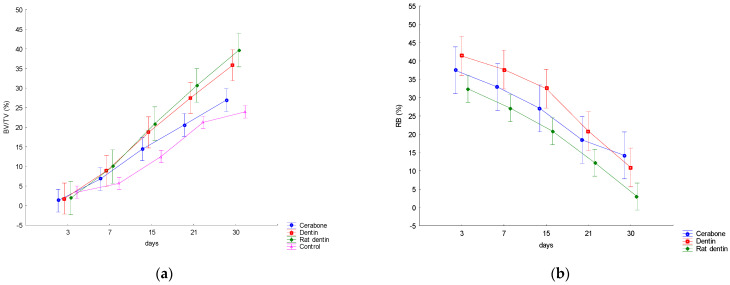
Dynamics of changes in histomorphometric parameters: (**a**) bone volume (BV/TV, %); (**b**) residual biomaterial (RB, %).

**Figure 3 materials-16-01600-f003:**
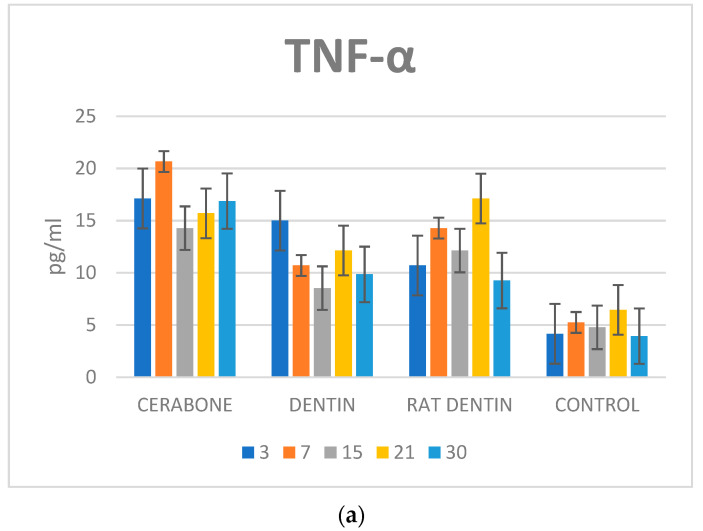
Dynamics of changes in serum concentrations of: (**a**) inflammatory (TNF-α, pg/mL), (**b**) osteogenic (OPN, ng/mL),and (**c**) vascular (VEGF, pg/mL)cytokines.

**Table 1 materials-16-01600-t001:** Experimental design.

Group Number	Group	Number of Animals (N)	Time Points (TP/days)	Total
1	Cerabone	5	5 (3,7,15,21,30 days)	25
2	Dentin	5	5 (3,7,15,21,30 days)	25
3	Rat dentin	5	5 (3,7,15,21,30 days)	25
4	Control	5	5 (3,7,15,21,30 days)	25
5	Tooth donor	3	/	3
TOTAL				103

## Data Availability

The data presented in this study are available on request from the corresponding author. The data are not publicly available due to privacy and ethical restrictions.

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
