# Peer review of "Dynamics of CSBD Healing after Implementation of Dentin and Xenogeneic Bone Biomaterial"

_materials, 2023, doi:10.3390/ma16041600_

Round 1

Reviewer 1 Report

Many thanks for the paper submission. This is an interesting paper that evaluates the autologous dentin and xenogeneic graft effect with CBCT. Some modifications are required to proceed to publication.

1) In the abstract the authors present thei noteworthy research with numerous acronyms: these terms are not explained in the abstract, please do so before report the single acronyms.

2) please add at least two more keywords

3) in the introduction section at line 31 please add this citation (new former reference no.2) with regards to biomaterials used in guided bone regeneration

  Chisci G, Fredianelli L. Therapeutic Efficacy of Bromelain in Alveolar Ridge Preservation. Antibiotics (Basel). 2022 Nov 3;11(11):1542. doi: 10.3390/antibiotics11111542. PMID: 36358197; PMCID: PMC9687015.   4)at line 43 the authors report the noteworthy paper of Pang et al. that reports the use of BioOss: a comparative comment between this material and Cerabone should be reported in the introduction section,as the authors report a paper regarding the use of Cerabone but cite as referral the use of a different biomaterial.   5) in the method section the authors report the use of calvarial graft to proceed in their research. Althought this is an animal research, in oral and maxillofacial surgery the use of calvarial bone graft is not so common: please state in the discussion section at line 254 "...the use of calvarial bone graft is already used in oral and maxillofacial surgery for onlay bone block for large defect.."   please cite the following   Sassano P, Gennaro P, Chisci G, Gabriele G, Aboh IV, Mitro V, di Curzio P. Calvarial onlay graft and submental incision in treatment of atrophic edentulous mandibles: an approach to reduce postoperative complications. J Craniofac Surg. 2014;25(2):693-7. doi: 10.1097/SCS.0000000000000611. PMID: 24621726.    

Author Response

Dear reviewer, 

the response is in the attachment below. 

The Authors

Reviewer 2 Report

Dear Editor, 

This manuscript studied the  dynamics (on days 3, 7, 15, 21, and 30) of the CSBD healing after implantation of the bovine devitalized dentin, rat dentin, and xenogeneic bone biomaterial. This manuscript should carefully revised and it needs extensive revision:

-Introduction needs to be improved and use the following reference and also, find their similar articles on the internet and improve your introduction section; https://www.sciencedirect.com/science/article/pii/S2238785420316495 (https://doi.org/10.1016/j.jmrt.2020.08.042)

-Method and materials should be more explained 

-Figure 1 should be in results not in method and materials section. Authors should explain the histopathological of figure 1 in result section and must be explained more 

-Results section is so poor and author should explain more 

-Discussion section is poor too, I suggest to authors to find similar article of mentioned above articles and also, find their similar articles and use them (2018-2023) in their discussion section.

Good luck, 

Author Response

(The authors gave the same response as above.)

Round 2

Reviewer 1 Report

 Many thanks for the revision performed. The paper needs in the discussion section a statement that in oral and maxillofacial surgery the use of calvarial bone graft is not so common: please state in the discussion section at line 378 "...the use of calvarial bone graft is already used in oral and maxillofacial surgery for onlay bone block for large defect.."  

please cite the following  

Sassano P, Gennaro P, Chisci G, Gabriele G, Aboh IV, Mitro V, di Curzio P. Calvarial onlay graft and submental incision in treatment of atrophic edentulous mandibles: an approach to reduce postoperative complications. J Craniofac Surg. 2014;25(2):693-7. doi: 10.1097/SCS.0000000000000611. PMID: 24621726.    

Author Response

Dear Editor,

the responses to the second round of revision are presented in the attached document.

Best regards,

The Authors

Reviewer 2 Report

Dear Editor,

The revised manuscript is acceptable.

Sincerely, 

Author Response

Thank you very much for your revision.

Best regards,

The Authors